# Prospects of Using Biocatalysis for the Synthesis and Modification of Polymers

**DOI:** 10.3390/molecules26092750

**Published:** 2021-05-07

**Authors:** Maksim Nikulin, Vytas Švedas

**Affiliations:** 1Belozersky Institute of Physicochemical Biology, Lomonosov Moscow State University, Lenin Hills 1, bldg. 40, 119991 Moscow, Russia; nikulin000@mail.ru; 2Faculty of Bioengineering and Bioinformatics, Lomonosov Moscow State University, Lenin Hills 1, bldg. 73, 119991 Moscow, Russia; 3Research Computing Center, Lomonosov Moscow State University, Lenin Hills 1, bldg. 4, 119991 Moscow, Russia

**Keywords:** biocatalysis, biodegradable polymers, enzymatic polymerization, biocompatible polymers, biobased polymers, biocatalytic monomer synthesis

## Abstract

Trends in the dynamically developing application of biocatalysis for the synthesis and modification of polymers over the past 5 years are considered, with an emphasis on the production of biodegradable, biocompatible and functional polymeric materials oriented to medical applications. The possibilities of using enzymes not only as catalysts for polymerization but also for the preparation of monomers for polymerization or oligomers for block copolymerization are considered. Special attention is paid to the prospects and existing limitations of biocatalytic production of new synthetic biopolymers based on natural compounds and monomers from biomass, which can lead to a huge variety of functional biomaterials. The existing experience and perspectives for the integration of bio- and chemocatalysis in this area are discussed.

## 1. Introduction

Recently, we have been witnessing an increasing use of enzymes for solving practically important problems, ranging from daily use, for example, in cosmetics, household care products and washing powders, to large-scale industrial processes. The attractiveness of enzymes is undoubtedly due to their unique properties as catalysts: high specificity, ability to work in mild conditions and, ultimately, environmental safety of use. However, it is necessary to note another factor that undoubtedly plays an important role in the development of enzyme technologies—the availability of enzymes. Nowadays, thanks to the techniques of genetic engineering, enzymes have become inexpensive and affordable catalysts even for large-scale applications. If we add to this the possibility of obtaining immobilized enzyme preparations, which, for example, provide hundreds of cycles in the batch stirred tank reactors and also retain their stability and activity for months when operating in continuous-flow biochemical reactors, it becomes clear that in many cases the cost of the biocatalyst does not contribute much to the cost of the product produced using the enzyme. “Enzymes are expensive” is an idea that is becoming obsolete [1]. In this regard, the use of enzymes is continuously expanding and spreading to new areas, even if it concerns the practically important transformations of nonspecific substrates and relatively low rates for enzyme-catalyzed reactions. Moreover, the successful development of methods of directed evolution [2,3,4], rational protein engineering based on bioinformatics analysis and molecular modeling provide a good chance to create more efficient variants of the wild-type enzyme [5,6,7,8,9]. The next step on the agenda is the development of methods for constructing de novo biocatalysts to carry out previously unknown reactions [10,11].

Important for an optimistic assessment of the prospects for the practical use of biocatalysis in new fields is the fact that many enzymes are able to efficiently convert not only their “natural” substrates, but also a wide range of synthetic organic compounds [12,13,14,15]. One of the areas of chemistry in which the possibility of using enzymes has been actively studied in recent decades is polymer chemistry. Several areas should be mentioned here. Thus, biocatalytic technologies are very actively developed and used in the textile industry in the processing of natural fibers, modification, dyeing and processing of biopolymers; these approaches are highlighted in a number of recent publications [16,17,18,19,20,21,22,23,24,25] and are beyond the scope of this manuscript.

The purpose of this review is to consider the possibilities of using enzymes as catalysts for polymerization, as well as obtaining monomers and oligomers for the synthesis of previously unknown, including biodegradable, polymers. This very dynamically developing field of chemistry is attracting interest due to the need for biodegradable and biocompatible polymeric materials with various functional properties for use in various fields of human activities, including medical applications [26,27,28,29,30,31]. Biosynthesis of natural biopolymers—proteins, polysaccharides, DNA and others—is not considered. The focus is on the results published in the last five years and devoted to the study of the possibilities of using enzymes for the preparation and modification of polymers not present in living organisms. The goal is to obtain a better understanding of what products can be obtained using enzymes, how they can be obtained and on what scale biocatalysis can be used in polymer chemistry. The relevance of the problem is due to the need for new polymeric materials and technologies for their production, especially having in mind the accumulation of waste in the life cycle of widely produced and used plastics from fossil raw materials that cause not yet fully recognized global environmental problems.

Initially, the term “enzymatic polymerization” was proposed to denote the chemical synthesis of polymers in vitro by a non-biosynthetic (nonmetabolic) pathway using an isolated enzyme [32]. Thus, we were talking about polymerization reactions catalyzed by enzymes in their native or immobilized forms. Historically, the first biocatalytic syntheses of polymeric and oligomeric compounds were carried out to produce small peptides using proteolytic enzymes at different stages of oligopeptide synthesis [33], as well as oligoesters formed by different dicarboxylic acids and diols applying lipases [34]. The demonstration that some enzymes, in particular lipases, can catalyze the conversion of hydrophobic compounds in nonaqueous solvents even more efficiently than in an aqueous medium [35,36] significantly stimulated research on the use of biocatalysis in organic synthesis as well as polymer synthesis. As a result, over the past thirty years, there has been a steady increase in the number of publications related to the use of enzymes for polymer chemistry (Figure 1).

About 3000 articles on the use of enzymes for the synthesis and modification of polymers, published over the past decade, testify to the active work of various scientific teams and the wide geography of relevant research. A number of reviews can be distinguished [32,37,38,39,40,41,42,43], in which information on enzymes and polymers that can be obtained with their help is considered. The main focus was on the lipase-catalyzed synthesis of biodegradable polyesters and polyamides, while interest in the use of other enzymes and biocatalytic production of monomers, as well as in the synthesis of other polymers, remained in the background and emerged later. Along with this, work on the use of enzymes to modify polymers deserves more attention. As a result, there are only a few relevant brief reviews in the literature [44,45]. Taking this into account, we tried in this review to consider the latest experimental work on enzymatic polymerization and modification of polymers, as well as the related problems arising in the course of biocatalytic polymerization reactions and ways to solve them, in order to assess the advantages and limitations of the use of enzymes for synthesis and modification of polymers.

## 2. Biodegradability of Polymers

With the awareness of the problem of accumulating plastic waste, the polymers used began to be classified into four categories, taking into account their biodegradability and the feedstock for their production (Table 1) [46]. Plastics can be produced either from renewable resources and the monomers of plant, microbial or animal nature, or from fossil resources—oil, gas and coal. They, in turn, can be biodegradable, that is, capable of relatively rapid complete decomposition into simple chemical compounds, for example, by naturally occurring microorganisms, or non-biodegradable, primarily destroyed in the natural environment under the influence of physical factors such as temperature and sunlight irradiation, forming micro and nanoparticles, and extremely slowly decomposed chemically by living systems or naturally occurring chemical reagents such as oxygen and water. Despite this simplified characterization of plastics, biodegradability is not obligatorily related to the feedstock for their production; it probably also should not be considered as a once and forever Corrected feature of any polymeric material that can be quantitatively evaluated for specified operating conditions.

Polyethylene, polypropylene, polystyrene and polyethylene terephthalate, which have played an important role in modern life, are typical petroleum-based plastics. Most of the plastics of this type are considered as non-biodegradable; however, in the presence of ester bonds in the polymer structure that can be hydrolyzed by esterases, some “petroleum” plastics can become biodegradable, such as poly-ε-caprolactone, poly(butylene succinate-adipate), poly(butylene adipate-terephthalate) and polyurethanes [47,48]. Bioplastics that are produced from renewable carbon sources and contain components extracted from plant biomass, for example, starch, cellulose, vegetable oil, lignin from plant or wood pulp, obtained by photosynthesis from atmospheric carbon dioxide, as well as products of microbial and animal nature all are expected to be biodegradable. Nevertheless, at least partially it is a misconception that depends on how we define biodegrability. Ideally, biodegradable plastic is defined as a material that is completely and within a reasonable time degraded to “simple” end-products such as carbon dioxide, water and ammonia by naturally occurring microorganisms such as bacteria, fungi and algae and does not leave any visually distinguishable or toxic residues [49]. Quantitative characterization of reaction rate and degree of conversion as well as product composition under defined degradation conditions should be used in the future to evaluate the degradability of plastics.

Polysaccharides with a high degree of substitution, such as cellulose acetate, used in the manufacture of photographic films, do not degrade in the natural environment. In addition, bio-polyethylene and bio-polyethylene terephthalate also cannot be considered biodegradable, although they can be obtained from bioethanol produced by fermentation of glucose and bioethyleneglycol, respectively. However, recently there has been a growing interest in the development of technologies for poly(ethylene terephthalate) depolymerization aiming at monomer recycling, and in this context, enzyme-catalyzed hydrolysis has been recognized as a promising alternative [50,51,52,53,54,55]. Therefore, the non-biodegradability of poly(ethylene terephthalate) may be reassessed in the nearest future.

Taking into account the reversibility of enzymatic reactions, it can be assumed that the products of enzymatic polymerization can be biodegradable plastics, and monomers for enzymatic polymerization can be obtained from both renewable and fossil resources. With the depletion of fossil resources and large-scale pollution of the planet by non-biodegradable plastic waste, the trend of sustainable exploitation of renewable resources for the production of biodegradable polymeric materials is growing [56,57]. Taking advantage of the structural diversity of various biomass resources, recent research activities were focused on the synthesis of renewable monomers and biodegradable polymers. Not only the use of renewable biomass, but also the development of ecologically friendly chemical and biocatalytic technologies is considered as a critical aspect for the sustainable production of materials based on derivatives of vegetable oil, terpenes, lignin, carbohydrates and other renewable resources [58].

## 3. Enzymes and Polymers Produced with Them

This section discusses different types of polymers that can be obtained using enzymes of different classes. Oxidoreductases can be used to obtain polyphenols and polyanilines by oxidizing the corresponding monomers. Polysaccharides and polyesters can be obtained using transferases and hydrolases. Hydrolases can be also be applied to synthesize polycarbonates, polyamino acids and polyesteramides, as well as their special case, polydepsipeptides. It should be noted that among other enzymes, lipases are most often used for the biocatalytic synthesis of polymers. Enzymes of other classes (lyases, isomerases, ligases) are still rarely used, mainly for the synthesis of highly specialized polymers and oligomers. These materials with relevant references are summarized in Table 2.

### 3.1. Oxidoreductases

Considering enzymatic polymerization with the participation of oxidoreductases, one should bear in mind radical polymerization, when an enzyme oxidizes a monomer to form radicals. Strictly speaking, the enzyme does not catalyze the polymerization reaction as such, but a reaction where active forms of the monomer are formed, i.e., the enzyme serves as the initiator [121]. The use of oxidoreductases is particularly well suited for in situ polymerization directly at the site of polymer application. This method can be promising for the production of biomedical adhesives and applications in various electrochemical and biosensors [122].

#### 3.1.1. Laccases

Laccases catalyze the oxidation of organic substrates, mainly derivatives of phenol and aniline, with molecular oxygen. The catalytic center of the laccase includes four copper atoms that form complexes with the imidazole groups of histidines and constitute an electron transfer chain. One copper atom is directly involved in the oxidation of the organic reducing agent, while the other three form a center where oxygen is reduced to water. Laccases from various sources are widely used for the synthesis of electrically conductive polyaniline with various counterions—working coatings for chemo- and biosensors and electrochemical devices [59,60,61,62,123], as well as for modifying cellulose with a polycatechol chain [63].

Recent work on the use of laccase from the fungus *Trametes versicolor* in polymer chemistry was devoted to the preparation of polydopamine films for surface modification (Figure 2) [64]. Comparison of the chemical and enzymatic method showed that the polymer obtained by the biocatalytic synthesis turned out to be more uniform and stable compared to the product obtained without using an enzyme. Several types of monomeric units were found in the structure of polydopamine, formed during the oxidation of dopamine by molecular oxygen in the presence of laccase. The reactivity of the terminal amino groups of dopamine residues was retained, which led to the formation of additional ester bonds between dopamine units, while this was not observed in the film obtained by the chemical method.

A high-redox potential laccase from the fungus *Trametes hirsuta* was used as a biocatalyst to synthesize poly(3,4-ethylenedioxythiophene), which is a promising material possessing both electronic and optical properties. This enzyme only requires molecular oxygen as an oxidant, which represents a great advantage for oxidative coupling reactions. The enzymatic oxidative polymerization of 3,4-ethylenedioxythiophene was performed using water-soluble sodium polystyrene sulfonate (PSS) or natural DNA as a template (Figure 3) [65,66]. The synthesized electrically conducting biocompatible poly(3,4-ethylenedioxythiophene)/DNA interpolymer complex can potentially be used for various biomedical purposes. Eco-friendly enzymatic synthesis of a promising electrode material for high-performance supercapacitors based on poly(3,4-ethylenedioxythiophene) and multiwalled carbon nanotubes doped with the redox-active compound sodium 1,2-naphthoquinone-4-sulfonate has also been reported [67].

#### 3.1.2. Peroxidases

Peroxidases catalyze the oxidation of a variety of organic substrates with hydrogen peroxide. The catalytic site of peroxidase includes a heme with an iron atom, which has a variable oxidation state. In the catalytic cycle, the Fe(IV) species are formed in the active site and oxidize the organic substrate to polymerizing active radicals. The use of peroxidases and other oxidoreductases for the polymerization of phenolic compounds is a promising alternative to chemical methods; however, these reactions are influenced by many factors (temperature, pH, solvent, enzyme origin, substrate structure), which often lead to dual effects. For example, the use of organic solvents, on the one hand, can increase the solubility of substrates, and, on the other hand, can suppress the activity of the enzyme or even lead to denaturation of the biocatalyst [124].

Phenol polymerization catalyzed by horseradish peroxidase (HRP) in the presence of hydrogen peroxide was efficiently carried out in an aqueous micellar system. Due to the effect of micelle solubilization, a phenolic polymer of moderate molecular weight was obtained in a high yield. The yield and molecular weight of the obtained polyphenol increased with a growing concentration of micelle-forming sodium dodecylbenzenesulfonate due to solubilization, and the authors managed to achieve a yield of 95% and an Mw of 12.6 kDa. The resulting phenolic polymer, consisting of a mixture of phenylene and oxyphenylene units, exhibited limited solubility and high thermal stability. Due to the absence of organic solvents, the method was considered as an environmentally friendly system for the production of phenolic polymers [68]. Polyphenols may be analogs of phenol formaldehyde resins widely used in industry, which raise concerns due to the toxicity of formaldehyde used in their production.

Peroxidase can also catalyze the polymerization of thiophene derivatives. It was found that environmentally friendly water-soluble poly(2-(3-thienyl)-ethoxy-4-butyl sulfonate) doped with titanium dioxide is an appropriate material for the creation of hybrid solar cells. Due to the presence of a polar sulfo group, the polymer is soluble in water and alcohol, which makes it possible to apply the polymer layer applying controlled evaporation. The synthesis of polythiophene polymer is carried out by the polymerization of monomers in an aqueous medium in the presence of hydrogen peroxide and peroxidase (Figure 4) [69]. Typically, in such reactions, hydrogen peroxide is added in small portions throughout the entire process to avoid the inactivation of the enzyme.

Various functional materials based on polymer particles can be obtained by oxidative polymerization of vinyl monomers, for example, acrylates, methacrylates, acrylamide and styrene, using horseradish peroxidase in microemulsions. This method makes it possible to form polymer particles stabilized by a surfactant and can be used to obtain polymer materials of the core–shell type, for example, with incorporated metal ions for use in optoelectronics [125,126]. Microbial catalase-peroxidase from *Escherichia coli* possesses an activity similar to horseradish peroxidase; it was applied in attempts to obtain oligomers of phenol, 3-methoxyphenol, catechol and aniline [127].

#### 3.1.3. Other Oxidoreductases

The Baeyer–Villiger oxidation of ketones by monooxygenases is attractive as an alternative method for the preparation of lactones, which are monomers in the synthesis of polyesters [128,129]. In contrast to the classical chemical method, where hydrogen peroxide and a catalytic amount of carboxylic acid are used to oxidize ketones, the biocatalytic method is advantageous due to the enantio- and regioselectivity, as well as reaction conditions, since the second substrate of monooxygenases is molecular oxygen. The enzymatic method may be more preferable when obtaining enantiomerically pure lactones of a complex structure. Cyclohexanone monooxygenase from *Thermocrispum municipale* (TmCHMO) has been successfully used for the oxidation of 3,3,5-trimethylcyclohexanone [70]. The resulting lactone can be subjected to enzymatic polymerization or polymerized chemically. The beneficial application of Baeyer–Villiger monooxygenases was demonstrated in the preparation of lactones from terpene derivatives, where the use of the classical approach with hydrogen peroxide would be impossible due to the formation of a significant amount of by-products [71,73]. The limiting factor when using this type of monooxygenases is the need to regenerate the coenzyme NAD(P)H. To solve this problem, NAD(P)H cyclic regeneration systems based on alcohol dehydrogenase or glucose oxidase with appropriate substrates were used together with monooxygenase [70,72].

Glucose oxidase can be used to generate hydrogen peroxide in situ, which in turn can initiate polymerization [130]. The combined use of glucose oxidase and peroxidase makes it possible to activate the monomer without the addition of hydrogen peroxide. In the glucose oxidase-catalyzed glucose oxidation by molecular oxygen, hydrogen peroxide is formed in situ and serves as a substrate of peroxidase, which oxidizes, for example, acetylacetone to initiate polymerization of polyethylene glycol methacrylate, which is used in the preparation of nanocomposite hydrogels [131].

### 3.2. Transferases

Transferases catalyzing the transfer of a group from a donor compound to an acceptor are most actively used in the preparation of hyaluronic acid derivatives, functional oligo- and polysaccharides that form glycogen-based self-healing hydrogels with ultra-stretchable, flexible and enhanced mechanical properties; supramolecular complexes; and conjugates for biomedicine, tissue engineering and modeling the properties of biopolymers [132,133,134,135,136]. Among the transferases, it is worth highlighting glycosyl and acyltransferases, which catalyze the synthesis of polysaccharides and polyesters, respectively. Polymerization reactions catalyzed by transferases are often analogous to biosynthetic pathways in vivo.

A novel synthetic strategy to produce homogeneous chondroitin polymers was suggested by combining stepwise oligosaccharides synthesis catalyzed by chondroitin synthase from *Pasterella multocida* with one-pot chondroitin chain polymerization [74]. It was found that, as for many synthases, a trisaccharide primer required to start the polymerization process is synthesized from uridine diphosphate derivatives of glucuronic acid (UDP-GlcA) and N-acetyl-galactosamine (UDP-GalNAc) by the same chondroitin synthase (Figure 5). This methodology demonstrated a strong relationship between the final sugar chain length and the molar ratios of reaction substrates. As a result, it allowed the synthesis of homogeneous chondroitin polymers with unprecedented narrow molecular weight distribution. Using this strategy, the unnatural zwitterionic and N-sulfonated chondroitin polymers can be synthesized by the incorporation of sugar nucleotide derivatives. Such polymers can be useful in modeling numerous pathological processes involving structurally defined chondroitin polymers.

Polyhydroxybutyrate synthase is an acyltransferase, the possibilities of which are studied in the context of the synthesis of poly-3-hydroxybutyrate (PHB), the most common member of the polyhydroxyalkanoate family [137,138,139]. Methods of microbiological synthesis of polyhydroxyalkanoates have become widespread, since polymers of this type are often used by bacteria as an intracellular source of carbon and energy [140,141,142]. There are also some recent studies on the synthesis of PHB from 3-hydroxybutyryl-CoA catalyzed by polyhydroxybutyrate synthase [75] and accompanied by propionyl-CoA transferase for regeneration of expensive coenzyme A [76]. PHB is similar to polypropylene and, thanks to the successful combination of biocompatibility and mechanical properties, potentially finds application in innovative biomedical systems and as a material for household use, degradable by soil bacteria without residue [143].

### 3.3. Hydrolases

Hydrolases are actively exploited by different industries and vigorously studied for application in polymer chemistry. The hot topic of the last decade—biocatalytic degradation of polymers to solve the still-underestimated problem of plastic waste—is outside the scope of this manuscript. Our goal is to evaluate the potential role and value of biocatalysis in the preparation of ecologically safe, biodegradable, biocompatible and functionally diverse polymers for different applications. Mechanistic studies of hydrolases considered as available model enzymes to study the power of biocatalysis from 1960s have demonstrated that numerous enzymes of this family are capable of not only catalyzing hydrolysis of substrates forming reactive acyl-enzyme intermediates but also transferring an acyl group to a nucleophile instead of water, thus performing synthesis. Such synthetic reactions cannot effectively proceed in vivo at low concentrations of substrates; however, they have been observed in vitro and can lead to high yields of synthesis at much higher concentrations of reagents that are typical for living organisms. Taking into account the reversibility of enzymatic reactions and selecting specific conditions, it becomes possible to use hydrolases as catalysts for reverse hydrolysis of synthesis reactions. Among hydrolases, the greatest attention is attracted mainly by esterases, including lipases and cutinases, as well as proteases. Using these enzymes, various classes of biodegradable functional polymers can be obtained, including polyesters, polyamides and polyesteramides [42].

#### 3.3.1. Lipases

Lipases belong to the family of α/β-hydrolases that form a typical α/β-hydrolase motif in the tertiary structure, which is an α-sandwich with parallel β-chains and one anti-parallel β-chain (Figure 6a). The active site of the lipase contains the catalytic triad Ser105-His224-Asp187, as well as the oxyanionic hole formed by the amino groups of Gln106 and Thr40, and is similar to serine proteases (Figure 6b). The catalytic mechanism of lipase, similar to that of serine proteases, includes the activation of the hydroxyl group of serine by relay charge transfer accompanied by a nucleophilic attack on the carbonyl carbon atom of the substrate, and the formation of an intermediate acyl enzyme with the release of a leaving group that is followed by a nucleophilic attack of water or added external nucleophile on the acyl enzyme with the formation of reaction products [45].

It should be noted that in industrial biotechnology, immobilized forms of biocatalysts are more often used, since the free enzyme can be rapidly inactivated under the conditions of synthesis, and it is also often difficult to separate the enzyme from the reaction mixture after the completion of the reaction in order to keep it active to be used repeatedly. One of the most common forms of biocatalyst is a lipase B preparation from *Candida antarctica*, known under the trade name Novozym 435. Novozym 435 (N435) is a commercially available immobilized lipase produced by Novozymes. It is based on immobilization via interfacial activation of lipase B from *Candida antarctica* on a resin, Lewatit VP OC 1600. This resin is a macroporous support formed by poly(methyl methacrylate) crosslinked with divinylbenzene. N435 is perhaps the most widely used commercial biocatalyst in both academy and industry. This preparation is granules with a diameter of 0.3–0.9 mm, containing 20% protein by weight and 1–2% water bound to the protein and necessary for its functioning. After the polymerization reaction, the granules are easily filtered or centrifuged and can be used repeatedly [40,144].

Lipases can catalyze a very wide range of reactions, covering not only the formation and hydrolysis of an ester bond, such as esterification or transesterification between an ester and an alcohol or an acid, as well as between two esters, but also amidation, aminolysis, aldol concentration, Michael addition, opening of lactone rings, epoxidation, etc. [45,145,146,147,148,149,150]. Lipases can exhibit enzymatic activity in various media, including acidic, neutral and alkaline aqueous; aqueous–organic solutions; organic solvents; or ionic liquids, and in the presence of detergents [151]. Thus, lipase is an enzyme with a unique catalytic potential activity that has been successfully used to obtain biodegradable polymers of many types.

##### Polyesters

Polyesters are polymers of hydroxy acids with hydrolytically labile ester bonds in the polymer chain used as thermoplastic materials. Due to their synthetic versatility and diverse properties, combined with regulatory compliance, polyesters have become the most widely studied synthetic polymers for biomedical applications [152]. Although all polyesters can be degraded by enzymatic or noncatalytic hydrolysis of ester bonds, aliphatic polyesters with a relatively short chain length are capable of degrading over a period of time required for most medical applications [153,154]. Among the polyesters, a special position in the context of biomedical application is occupied by polylactides and polyglycolides, as well as their copolymers. Polymers of this type are of primary importance in regenerative medicine. Materials based on polylactides and polyglycolides can be used in surgery and in vascular and tissue engineering [155,156,157]. Other representatives of polyesters, for example, poly-ε-caprolactone and poly(butylene succinate), due to their suitable thermomechanical properties are also attractive as biomedical materials [158,159]; however, despite the high biocompatibility of polyesters, the possible side effects of their use in medicine should be taken into account, for example, a local decrease in pH upon their degradation [160].

The most common methods for the preparation of polyesters include lactone ring-opening polymerization as well as polycondensation of hydroxy acid derivatives or dicarboxylic acid derivatives with diols (Figure 7).

Traditional chemical catalysts in the synthesis of polyesters have led to the production of valuable commercial products. However, in this case, polymerization usually takes place at high temperatures (150–280 °C) using complex organic catalysts [161,162,163,164] or catalysts containing metal ions, including heavy metals [165,166]. Chemical synthesis at high temperatures is often accompanied by side processes, with a significant limitation in this case being the difficulty in dealing with thermally or chemically unstable monomers. In addition, stringent requirements are imposed on the content of heavy metals for biomedical materials based on biodegradable polymers, and it becomes necessary therefore to thoroughly purify the resulting material from a chemical catalyst.

In the context of obtaining specific functional polyesters for biomedical applications, the most promising method is biocatalytic polymerization. Lipase is a nontoxic natural biocatalyst that provides mild conditions for the polymerization in combination with high chemo-, regio- and enantioselectivity, which is often difficult to achieve by traditional methods of chemical synthesis [167,168,169,170]. As already noted, the use of lipase is not limited to aqueous solutions; the enzyme can function in complex heterogeneous systems, including nonaqueous systems, which will be discussed below. In addition, lipase can be effectively used at polymerization of “renewable” monomers obtained by microbiological methods, such as adipic, itaconic, succinic, terephthalic and furandicarboxylic acids, as well as ethylene glycol, glycerol, 1,4-butanediol, 1,3-propanediol and sorbitol [171,172].

The putative catalytic mechanism of enzymatic polymerization reactions involving lipases includes the stage of formation of an enzyme-activated monomer, which further participates in the stages of chain initiation and growth (Figure 8). The nucleophilic attack of the carbonyl carbon atom of the enzyme-activated monomer with the terminal hydroxyl group of the chain leads to the elongation of the polymer chain by one monomeric unit [173]. A similar mechanism can be realized in other types of reactions for the preparation of polyesters, shown in Figure 7.

Among recent works on the enzymatic production of polyesters, one can highlight the production of biodegradable conjugates of proteins and poly-ε-caprolactone using ring-opening polymerization of ε-caprolactone catalyzed by lipase B from *Candida antarctica* [77], the production of polyesters from macrolactones, for example, ω-pentadecalactone, and its unsaturated derivatives, using Novozym 435 with a yield of 80-90% and a molecular weight of up to 130 kDa [78,174], as well as copolymers based on ε-caprolactone and ω-pentadecalactone [79], ε-caprolactone and 5-hydroxymethyl-2-furancarboxylic acid [80]. Polycondensation methods have been successfully used in the preparation of polyesters based on aromatic and aliphatic diols and dicarboxylic acids obtained from renewable raw materials using lipase B from *Candida antarctica* (CALB) [81,82,83], as well as in the preparation of polyesters containing, in addition to methylene fragments, simple thioether groups [84].

Focusing on an eco-friendly approach, biodegradable poly((butylene succinate)-co-(dilinoleic succinate)) copolymers were successfully synthesized via various processes and catalytic systems. In the first procedure, a two-step synthesis in diphenyl ether was performed using *Candida antarctica* lipase B as a biocatalyst. A second material was produced via two-step melt polycondensation in the presence of a heterogeneous titanium dioxide/silicon dioxide (C-94) catalyst. GPC analysis revealed M_n_ to be 51 kDa for C-94-catalyzed multiblock copolyester and 25 kDa for CALB-catalyzed material. However, the degree of crystallinity was lower for polymer produced with the heterogeneous catalyst [85].

##### Polyamides

Polyamides are considered as one of the most widely used types of polymers in various industries, medicine and the production of a wide range of consumer goods. The vast majority of polyamides, mainly nylon, are petroleum-based. To obtain functional polyamides of a complex structure, for example, for biomedical purposes, the use of enzymatic polymerization seems to be expedient. As in the case of polyesters (Figure 7), polyamides can be obtained by polycondensation of amino acids and their derivatives, polycondensation of dicarboxylic acid derivatives with diamines and ring-opening polymerization reactions with lactam by lipase-catalyzed transformations, both in the presence of solvent and without it [41,42,102].

Polyamides, which can be obtained from renewable raw materials, are of great interest. For example, using CALB, oligomers were synthesized based on a monomer obtained by radical addition of cysteamine, which is a metabolite of coenzyme A, to methyl oleate (Figure 9). Polycondensation was carried out at 80–140 °C without solvent, while the authors reported that polymerization in the presence of lipase was more efficient than with the organic catalyst 1,5,7-triazabicyclo[4.4.0]dec-5-ene; the conversion reached 90% in less than 1 h [103].

Polyamides based on 2,5-furandicarboxylic acid containing aromatic fragments can act as an alternative to petroleum-based polyamides and be used in the production of engineering thermoplastics and high-tech materials. Enzymatic polycondensation of dimethyl-2,5-furandicarboxylate and aliphatic diamines of various lengths (from 4 to 12 carbon atoms) was carried out in toluene at 90 °C using Novozym 435; polyamides with a molecular weight of up to 48.3 kDa and a yield of about 50% were obtained [104].

Oligomeric β-peptone-peptidomimetics with a length of 5-6 units based on ethyl ester of N-(2-hydroxyethyl)-β-alanine, which is a secondary amine, were obtained using immobilized CALB with 60% yield. Chemical modification of oligomeric β-peptones with poly-ε-caprolactone led to the formation of biodegradable films for biomedical applications, particularly for cell adhesion during tissue culture [105].

Proteins are polyamino acids and form the basis of body tissues; therefore, biomaterials based on polyamino acids can be used as surgical suture materials, hemostatic agents and scaffolds for tissue engineering. Side groups of amino acids can be further modified, for example, by attaching a molecule of a physiologically active compound, and thus, polyamino acids can be used in drug delivery systems [175].

An interesting example of the use of polyamino acids in medicine is the development of micellar nanostructures based on polyaspartic acid for smart drug delivery systems. Enantiomerically pure polyaspartic acid (Figure 10) was obtained from diethyl aspartate by enzymatic polymerization without solvent using CALB at 80 °C for 24–72 h with a yield of about 40% and a degree of polymerization of about 30. It is a highly water-soluble ionic polymer containing up to 97% β-units (Figure 10) [106]. Materials based on polyaspartate are degraded in the body by lysosomal hydrolytic enzymes.

##### Polyesteramides

One of the main requirements for a biodegradable biomedical material is that the decomposition products of the implant must be nontoxic, metabolized in the body and excreted. The safety of products formed during the degradation of biomaterial is a serious problem, the solution of which raises interest in the development of polymeric materials that include natural metabolites as monomeric units. These materials include polyhydroxy acids, polyamino acids and polyesteramides based on hydroxy and amino acids, which are also called polydepsipeptides. Polyhydroxy acids are the most studied type of biodegradable biomedical material. The use of polyglycolide and polylactide has been mentioned earlier. Polyamino acids, despite their recognized advantages, are used in medicine much less often than polyesters. This is due to their high crystallinity, low degradation rate and possible immune response.

Polyesteramides (polydepsipeptides) include amide and ester bonds in the main chain. Polyesteramides, as well as polyesters, can degrade at a sufficient rate due to chemical or enzymatic hydrolysis of the ester bond. On the other hand, polyesteramides, like polyamides, should have better thermomechanical properties due to the formation of hydrogen bonds between amide groups [176]. Thus, the combination of amide and ester bonds in one polymer presents additional opportunities for the creation of new materials with desirable properties for biomedical application [177,178,179,180,181]. Various derivatives of hydroxy and amino acids, dicarboxylic acids, cyclic depsipeptides, diols, diamines, amino alcohols, lactams and lactones can be used for the synthesis of polyesteramides [182,183,184,185].

Chemical methods for the synthesis of polyesteramides include thermal polycondensation of dicarboxylic acid derivatives, diols and amino acids in bulk; interfacial polymerization; polycondensation in aqueous solution; and ring-opening polymerization [182,186,187]. The field of application of chemical methods for the synthesis of polyesteramides is limited by a number of factors. High reaction temperatures during polymerization in bulk are undesirable in the case of thermolabile monomers, as well as when using enantiomerically pure compounds due to possible racemization. Working with polyfunctional compounds requires the inclusion of the steps of including and removing protective groups, which makes the synthesis multistage and costly. Polycondensation in aqueous solution is characterized by mild reaction conditions (temperature <80–90 °C, atmospheric pressure) and minimal side processes; however, it is necessary to take measures aimed at overcoming thermodynamic limitations and shifting equilibrium towards polymer formation. In the case of ring-opening polymerization, derivatives of 2,5-diketomorpholine are used as starting monomers; 2,5-diketomorpholine is a cyclodepsipeptide—a condensation product of amino and hydroxy acids [188]. The most common catalyst used for this reaction is tin(II) octoate (tin(II) 2-ethylhexanoate) This is a typical catalyst for ring-opening polymerization of lactones [189,190,191]. In this regard, it becomes necessary to thoroughly purify polymer products, especially those intended for biomedical purposes, from heavy metals that are part of organometallic catalysts.

By analogy with the reactions for the preparation of polyesters, lipases can catalyze ring-opening polymerization of 2,5-diketomorpholine derivatives. Biodegradable polyesteramide containing residues of glycine and L-lactic acid was obtained by ring-opening polymerization of 6-(S)-methyl-morpholine-2,5-dione using porcine pancreatic lipase at 100 °C for 6 days with a yield of 68% and a molecular weight of about 15 kDa [107]. The mechanism of this reaction is similar to that shown in Figure 8. This mechanism assumes the absence of a chain termination step, since, theoretically, one more monomeric unit can be attached to the OH-terminus of any formed polymer chain. However, it is obvious that the molecular weight cannot increase infinitely, since the viscosity of the reaction mixture increases with the growth of the polymer chain, and it introduces kinetic restrictions on the polymerization process.

Another method for the enzymatic production of polyesteramides is the copolymerization of ε-caprolactone and β-lactam or their derivatives using CALB in toluene for 3 days at 90 °C. The average degree of polymerization of the obtained polyesteramide was 29, and the yield reached 50% [108]. Biodegradable polyesteramide based on 6-hydroxyhexanoic and L-aspartic acids with a molecular weight of up to 13 kDa was obtained by the enzymatic polycondensation in toluene (Figure 11). Regioselective enzymatic polymerization provides the formation of a polymer containing 98% β-bonds [109].

The polymerization of 2,5-furandicarboxylic acid, one of the key building blocks for the preparation of furan polymers, is often accompanied by side reactions. Due to the mild reaction conditions, enzymatic polymerizations became an excellent candidate to address this issue [192,193]. A green and effective method to prepare different furanic–aliphatic polyesteramides was reported. Polymers with M_w_ up to 21 kDa were successfully synthesized by a Novozyme 435-catalyzed polycondensation of dimethyl 2,5-furandicarboxylate with aliphatic diols, diamines or amino alcohols, using toluene at 90 °C [110].

#### 3.3.2. Cutinase

Cutinases are members of the serine hydrolase family and include a typical catalytic triad Ser-His-Asp and an oxyanionic hole. The natural substrate for cutinases is cutin, which covers the surfaces of higher plants. It is a water-insoluble polyester composed of hydroxy and epoxy fatty acids derived from palmitic and stearic acids. Cutinases are capable of catalyzing ester hydrolysis, esterification and transesterification reactions. In this regard, cutinases are considered as an alternative to widespread lipases in the context of obtaining biodegradable polymers, in particular polyesters [111,112]. On the example of the polymerization reaction of dimethyladipate with 1,4-butanediol, it was shown that at 70 °C in the absence of solvent, cutinase 1 from *Thermobifida cellulosilytica* demonstrates better results than Novozym 435: using immobilized cutinase, the conversion per day increased from 78 up to 86%, and the molecular weight of the polymer product doubled and amounted to about 2 kDa. Bioinformatic analysis revealed that this may be due to the greater availability of the active center of cutinase because of its conformational features in both aqueous and hydrophobic media compared to CALB [113].

#### 3.3.3. Proteases

Proteolytic enzymes in the context of enzymatic polymerization are considered as catalysts in the reactions of synthesis of oligopeptides and oligopeptidomimetics with various functional and physical properties [194]. Among proteases, papain attracts the most attention, probably because of its wide substrate specificity, enzyme stability and ease of its production. As a rule, protease-catalyzed polymerization reactions are carried out in an aqueous medium at a pH of about 8, a temperature of about 40 °C for several hours, while oligomers are formed with a degree of polymerization not exceeding 10. Esters of amino acids act as monomers, and there is an increase in yield when using benzyl esters of amino acids [195].

Polypeptides are promising biodegradable polymers with unique physicochemical properties. However, due to strong intermolecular interactions, polypeptides do not possess thermoplasticity, and it is difficult for them to give the desired shape at elevated temperatures. The pathway of copolymerization of amino acids with nylon units using papain turned out to be a promising way, with the formation of a thermoplastic material (Figure 12) [114,115].

A crystalline block copolymer of L-lysine and L-alanine in an aqueous solution was obtained using papain with a yield of more than 40% within 30 min. The block copolymer has potential applications in tissue engineering and drug and gene delivery systems and has self-assembling properties [116]. A thermostable material with oxidation stability based on cysteine oligopeptide with a degree of polymerization of 8.8 was obtained in 78% yield in the presence of proteinase K in an aqueous solution with phosphate buffer at pH 8 [120].

When comparing papain and bromelain in the polymerization of L-phenylalanine methyl ester in an aqueous medium pH 8 at 40 °C for 3 h, it was shown that papain demonstrates the best results: the degree of polymerization of oligophenylalanine was 6.62, yield 72.9%; when using bromelain, the yield was only 16%, and the degree of polymerization was 5.2. Thus, papain can be used in the synthesis of biologically active oligopeptides [196].

A peptidomimetic based on alanine and unnatural 2-aminoisobutanoic acid was obtained using papain in an aqueous medium at 40 °C for 2 h with a yield of 30% and a molecular weight of 0.9–1.8 kDa [117]. Papain has also been used in the synthesis of oligopeptides to modify polylactide [118]. The synthesis of a zwitterionic polypeptide based on glycine and histidine, potentially important for the modification of cellulose in the paper industry, was carried out using papain in an aqueous medium at 40 °C for 2 h with a yield of 62% and a molecular weight of 2.4 kDa [119].

The enzymatic synthesis of methotrexate (MTX) catalyzed by α-chymotrypsin was studied for the first time. The proteolytic enzyme displayed activity for the synthesis of MTX oligomers composed of six repeating units (DP_avg_ = 1.5) [197]. The oligomerization of this drug would represent an alternative route for the treatment of many diseases by reducing some side effects associated with the monomer administration.

## 4. Biocatalytic Modification of Polymers

Naturally occurring polymers, such as polysaccharides and polypeptides, often have pendant functional groups in the polymer chain. The opportunity of modifying such groups makes it possible to expand the application field of natural polymers by changing the physicochemical properties of the modified material. The properties of synthetic polyamides, polyesters and polyesteramides can also be improved by surface modification as a result of partial hydrolysis of amide and ester bonds. Enzymes of different classes can be used for effective and selective modification and functionalization of polymers [45,198,199,200,201].

Modification of nanoparticles consisting of the chitosan–chondroitin sulfate complex by phosphorylation of 6-OH-groups was carried out using hexokinase (Figure 13). Modification with negatively charged phosphate groups improves the permeability of nanoparticles through the mucous membrane. The uptake by epithelial cells is facilitated due to a decrease in the zeta-potential due to partial hydrolysis of phosphate groups in the presence of alkaline phosphatase at the surface of epithelial cells. This system can be considered promising for biomedical applications, in particular for gene delivery [202].

Modification of the polylactide surface with cutinase led to partial hydrolysis of ester bonds; while the molecular weight remained practically unchanged, the material became more hydrophilic due to the presence of additional hydroxyl and carboxyl groups. This system is considered as a promising method for the implementation of controlled release of drugs. In particular, the interaction of modified films of polylactide with doxorubicin, a cationic chemotherapeutic anticancer drug, was studied [203].

Modification of cyanophycin, a peptide polymer composed of N-arginyl-aspartate units, using type II peptidyl arginine deiminase from *Oryctolagus cuniculus* led to the conversion of arginine residues to citrulline [204].

Enzymatic modification of polymers can also be considered as a promising way to improve the properties of materials produced by various industries. Surface modification with alkalase was carried out to improve the dyeing of synthetic polyamide fibers [205]. Hydrophobization of cotton by creating a polyester coating was carried out using lipase [206]. Laccase-catalyzed modification of chitosan with fatty aromatic acids, such as caffeic, gallic and cinnamic acids, led to an increase in the antibacterial activity of the chitosan-based material used in pharmaceuticals, cosmetology and food industry [207,208].

## 5. Biocatalysis Engineering

The technological implementation of scientific developments on the use of enzymes in the production of polymers is associated with the solution of a number of problems: the choice of the feedstock and the optimal form of the catalyst, the search for optimal conditions for the polymerization or polymer modification, the isolation of the target product, etc. In recent years, the concept of ‘biocatalysis engineering’ has been formed with a purpose of considering the enzymatic process as a whole, which entails engineering its different components: substrate engineering, medium engineering, protein (enzyme) engineering, biocatalyst (formulation) engineering and reactor engineering (see the summarizing conclusions in Figure 14; a more detailed discussion is provided in the following text) [209]. Let us try to look from this angle at the work on the studies of the possibilities of using enzymes in the field of polymer chemistry to which this review is devoted. There are many publications in the literature that describe certain biocatalytic transformations; however, in most cases, no justification for the selected reaction conditions or the optimization is given. However, for a seamless design of the entire process, first of all, it is necessary to optimize the reaction conditions, such as the presence of a solvent, temperature, component concentrations and other physicochemical parameters. There are only separate works that at least partially answer the question of the optimal conditions for carrying out the synthesis.

So, the efficiency of the *Thermomyces lanuginosus*-immobilized lipase for ring-opening co-oligomerization of ε-caprolactone and δ-gluconolactone was demonstrated and the optimal reaction conditions were determined by experimental design and statistical analysis. The enzymatic in vitro oligomerization process was optimized by a three-factorial/three-level experimental design, using the Box–Behnken method. The selected independent variables were the temperature, the enzyme amount and the molar ratio of monomers, while the co-oligomerization degree and the mass average molecular mass (calculated from MALDI-TOF MS data) were the response variables. The results indicate that temperature has the most significant effect and is directly correlated with the formation of linear co-oligoesters. The overall effect of the other variables was also significant [86]. Various parameters (type of lipases, temperature, pH, stirring type and rate and monomer carbon chain length) of the polycondensation in an oil-in-water (o/w) miniemulsion (>80% in water) were evaluated when enzymatic polycondensation of dicarboxylic acids and dialcohols in aqueous polymerization media using free and immobilized lipases was developed [87,88].

### 5.1. Enzyme Engineering and Biocatalyst Formulation

Another important aspect is the nature and form of the biocatalyst. Under laboratory conditions, the free enzyme is often used in the form of, for example, a lyophilized powder. However, on an industrial scale, the overwhelming majority of enzymatic processes are carried out using immobilized enzymes in order to simplify the separation of the catalyst from the reaction mixture at the end of the reaction, as well as to provide the possibility of multiple uses of the stable form of the biocatalyst. The used immobilized forms of biocatalysts are enzymes covalently or noncovalently bound to the substrate, cross-linked enzyme aggregates and enzymes incorporated into polymer carriers [210,211,212,213]. A widespread form of lipase B from *Candida antarctica* in the biotechnological industry is Novozym 435, used in numerous studies of enzymatic polymerization reactions which were discussed above [144]. Some effective biocatalytic conversions using multienzyme systems and methods of joint immobilization were successfully used, where all enzymes participating in the reaction were immobilized on inorganic nanoparticles [214].

Protein engineering strategies were applied to construct mutants with improved catalytic properties. A structure-based engineering strategy was used to redesign CALB active site in order to improve catalytic efficiency of poly(ε-caprolactone) synthesis and size of polymer products. From the constructed library containing 1410 single mutants, 8 improved mutants were identified, among which mutant I285R showed nearly 3-fold increased catalytic efficiency for the ε-caprolactone opening, the first step of the reaction, and allowed the synthesis of poly(ε-caprolactone) with 30% increased molar mass compared to parental wild-type CALB enzyme under the same reaction conditions [89]. In another work, the authors investigated which changes in the CALB structure can shift the catalytic equilibrium between esterification and hydrolysis towards polymerization. Two approaches were tested: (I) removing the glycosylation of CALB to increase the surface hydrophobicity and (II) introducing a hydrophobic lid adapted from *Pseudomonas cepacia* lipase to enhance the interaction of a growing polymer chain to the elongated lid helix. While the lid variants showed a minor positive effect on the polymerization activity, the deglycosylated CALB demonstrated a clearly reduced hydrolytic and enhanced polymerization activity [90].

PEGylation was explored as an efficient strategy to improve esterase’s catalytic performance. The authors PEGylated three esterases: CALB, lipase from *Thermomyces lanuginosus* (TL) and cutinase from *Fusarium solani pisi* and evaluated their catalytic performance by using the biosynthesis of poly(ethylene glutarate) as a model reaction. The data revealed a higher polymerase activity for the lipase TL and cutinase PEGylated forms. This may be due to a more open active site cavity for the PEGylated catalysts facilitating the catalysis [215].

Obviously, due to viscosity and diffusion problems, specific biocatalyst formulations would be useful to overcome these problems and facilitate the separation of the biocatalyst from the reaction mixture and the resulting products. So-called soluble–insoluble enzyme preparations appear to be promising candidates that can work in homogeneous solutions and then can be isolated from the reaction mixture in their insoluble form [216]. Various forms of such smart biocatalysts are being actively developed but are not yet used in polymer production [217].

### 5.2. Substrate Engineering

The synthesis of optically pure polymers is one of the most challenging tasks in polymer chemistry. Substrate engineering, i.e., structural modification of monomers by the introduction of different substituents, can be used to modulate biocatalytic conversion. A remarkable effect was observed in Novozym 435-catalyzed polycondensation between D-/L-aspartic acid diester and diols in an attempt to prepare helical chiral polyesters. Compared with D-Asp diesters, the fast-reacting L-Asp diesters easily reacted with diols to provide a series of chiral polyesters with specific helical structures containing N-substituted L-Asp repeating units. It was observed that the introduction of bulky N-acyl substituents, like N-Boc and N-Cbz groups, was more favorable for this polymerization than using small ones, probably due to the different binding and orientation of substrates in the active site of the enzyme. The slow-reacting D-Asp diesters were also successfully polymerized by modifying the substrate structure to create a “nonchiral” condensation environment artificially [91].

### 5.3. Biocatalytic Systems Engineering

A huge number of variable conditions for enzymatic polymerization reactions lead to the need to engineer a variety of biocatalytic systems used in the preparation of polymers. The unexpected effects of mechanical activation were shown when the reaction was carried out under the conditions of a ball mill. The authors have reported that the collision of balls during the operation of the mill can increase the activity and enantioselectivity of lipase B from *Candida antarctica* in immobilized and free forms [218]. Efficient oligomerization of L-amino acid esters with an almost quantitative yield and a degree of polymerization from 6 to 26 was also observed under the conditions of a ball mill in the presence of papain, demonstrating that in industrial production the ball mill can be replaced by a twin-screw extruder [219]. An improvement of the enzymatic polymerization can be achieved also by treating the reaction mixture with ultrasound [92], as well as in mini- and nanoemulsion systems [93,94].

The viscosity problem is not a trivial one in polymer chemistry. This aspect is especially acute when using biocatalytic reactions. In a course of polymerization with an increase in the molecular weight of the resulting polymeric product, the viscosity of the reaction mixture inevitably increases or the polymer precipitates, and then the system becomes heterogeneous. These factors limit the maximum molecular weight of the polymer obtained by enzymatic polymerization under the conditions of a pure monomer melt or its solution in a particular solvent, whereas an increase in temperature leads to inactivation of the biocatalyst. In addition, with an increased viscosity, the activity of the enzyme and the rate of polymer synthesis decrease [220,221].

The strategy of using thin-film processes at reduced pressure (about 70 mbar) using covalently immobilized lipases made it possible to bypass the viscosity problem in the preparation of polyesters based on adipic and itaconic acids and 1,4-butanediol by polycondensation at 50 °C. The technical implementation of the process consisted in the use of a rotary evaporator: the reaction was carried out in a thin film on the inner surface of a rotating flask under reduced pressure. The immobilized enzyme was separated by filtration from the mixture of liquid polymerization products and unreacted monomers. The results open up new prospects for expanding the applicability of biocatalysts in other viscous systems and solvent-free syntheses [95]. Another approach to circumvent the viscosity problem was to carry out enzymatic polymerization of ω-pentadecalactone in the presence of immobilized CALB by reactive extrusion under high shear and temperature conditions. Conversion of the monomer took place almost quantitatively in 15 min, and the molecular weight of polypentadecalactone was 163 kDa [96]. Novozym 435 granules were separated from the reaction mixture by filtration after dissolving the polymer product in xylene.

The strategy of carrying out biocatalytic, chemical and chemoenzymatic polymerization in flow-through microreactors has shown good prospects [222,223,224]. The design of the microreactor allows carrying out heterogeneous reactions in a continuous mode, in organic media and at elevated temperatures [225,226]. This approach proved to be convenient for the preparation of block copolymers by lipase-catalyzed ROP. The convenience of handling the copolymerization conditions and processes, reduced overall reaction time and well-controlled molecular weights and distributions were achieved by employing this microfluidic enzyme–organocatalysis combination strategy [227]. Similar microreactor-based platforms can readily be extended to other enzyme-based systems, for example, high-throughput screening of new enzymes, and to precision measurements of new processes where continuous flow mode is preferred. At the same time, the problem of scaling up this process should be addressed.

### 5.4. Solvent Engineering

There are numerous examples showing that enzyme-catalyzed reactions can be carried out in anhydrous organic solvents and solvents with different amounts of water, including two-phase systems [228,229]. Moreover, for many enzymatic synthesis reactions, water is not an ideal medium due to unfavorable thermodynamics; low solubility of hydrophobic monomers, oligomers and polymers; the possibility of microbial contamination; etc. An additional advantage of a nonaqueous medium in some cases may be the ease of separating and reusing enzymes and isolating products.

Some of the most biotechnologically exploited enzymes nowadays are lipases; the wide spectrum of reactions catalyzed by enzymes of this family was discussed in the previous sections. Hydrolysis of fatty acid triglycerides is the main function of lipases in vivo. Such hydrophobic substrates are insoluble in water and form micelles upon reaching the critical micelle concentration (CMC). Since lipase operates with two substrates, ester and water, it must work at the interface. Indeed, the activity of lipase in an aqueous medium increases sharply upon the addition of an organic phase. This phenomenon is called surface, or interfacial, activation, which is explained by a change in the conformation of the enzyme upon contact with the nonpolar phase [230]. X-ray diffraction data usually show a “closed” conformation in which a movable “cap” consisting of two α-helices blocks the active site. This “lid” opens when the enzyme is exposed to the water–organic interface, and the catalytic activity of the lipase increases [231].

In the previous sections, numerous examples of lipase-catalyzed reactions in pure organic solvents were shown, which can be characterized by the lipophilicity index logP (where the parameter P is defined as the ratio of the solubility of a given solvent in the organic and aqueous phases, namely in a two-phase octanol–water system). The best solvents for reactions catalyzed by lipases are considered to be those with a lipophilicity index greater than 1.9. The most commonly used solvents in laboratory studies are toluene, benzene, diphenyl ether, hexane and cyclohexane [232]. The possibilities of using “green” solvents, such as ionic liquids, particularly deep eutectic solvents, and superfluidic carbon dioxide, which will be discussed below, have also been shown. The use of “green” solvents is probably limited so far due to their high cost and the need for thorough cleaning, since the presence of impurities can seriously affect their physicochemical properties and enzyme activity.

#### 5.4.1. Ionic Liquids

Ionic liquids are organic salts with melting points below 100 °C. As a positively charged ion, various derivatives of quaternary ammonium cations are usually used, including heterocyclic ones, such as 1-butyl-3-methylimidazolium; examples of anions include tetrafluoroborate, trifluoromethyl acetate, bis((trifluoromethyl) sulfonyl)imide, hexafluorophosphate, trifluorophosphate, trifluoride sulfate and chloride [233,234]. The physicochemical properties of ionic liquids strongly depend on the nature of the ions included in the composition. The number of possible combinations of cations and anions is very large, which makes it possible to create a suitable ionic liquid for specific reaction conditions. In recent years, ionic liquids have increasingly attracted attention as solvents for the biocatalytic production of polyesters by the reaction of lactone ring-opening polymerization. In some cases, ionic liquids have been found to improve the solubility of substrates or products and also provide higher enzyme stability and activity. For example, polycaprolactone with a molecular weight of about 6 kDa and a yield of 85% was obtained under the conditions of the Novozym 435-catalyzed polymerization of ε-caprolactone in 1-butyl-3-methylimidazolium tetrafluoroborate at 60 °C for 2 days. In this case, the resulting polymer had a clear spherulite structure, which can provide a large surface area for cell adhesion [97]. Using an ionic liquid with a reduced viscosity based on a tetraalkylphosphonium cation modified with oligoethylene glycol and an anion of bis((trifluoromethyl)sulfonyl)imide in the presence of Novozym 435, it was possible to obtain polylactide and polycaprolactone with molecular weights of about 20 kDa [98]. Thus, ionic liquids seem to be quite attractive reaction media for enzymatic polymerization.

#### 5.4.2. Deep Eutectic Solvents

Deep eutectic solvents (DES) are an eye-catching class of solvents formed by mixing organic ammonium salts and hydrogen bond donor compounds (for example, urea, glycerol, ethylene glycol, thiourea, acetamide, benzamide and malonic and oxalic acids). In these solvents, the hydrogen bond donor interacts with the anion (Figure 15), causing a decrease in the melting point of the mixture. One of the most widespread deep eutectic solvents is a mixture of urea (m.p. 133 °C) and choline chloride ((2-hydroxyethyl)-trimethylammonium, m.p. 247 °C) in a molar ratio of 2:1, having a melting point of about 12 °C [235]. Like ionic liquids, deep eutectic solvents have low volatility and high thermal stability; however, unlike most ionic liquids, such solvents are biodegradable, nontoxic, very easy to prepare and have great potential as “green” solvents [236].

To test whether deep eutectic solvents can be a suitable medium for reactions catalyzed, for example, by lipase, the authors of one of the studies used a model transesterification reaction of the vinyl laurate with alcohols of different hydrocarbon chain lengths (Figure 16) [237]. Various combinations of choline chloride, ethylammonium chloride, as well as ethylene glycol, glycerol, urea and malonic and oxalic acids were used to prepare deep eutectic solvents. In mixtures of choline chloride/urea (1:2) and choline chloride/glycerol (1:2), for 16 h of reaction, the degree of conversion approached 100%, while in other combinations of components the degree of conversion was from 3 to 43%. It was noted that the high viscosity of dicarboxylic acid solvents makes mixing difficult and creates diffusion restrictions.

A wide range of reactions of enzymatic and chemoenzymatic synthesis, including transesterification, epoxidation and formation of C-C bonds, laccase-catalyzed oxidative polymerization carried out in deep eutectic solvents has been considered in the literature [238,239,240]. It should be noted that some substrates of biocatalytic conversions can serve as components of deep eutectic solvents, providing an additional parameter for optimization.

#### 5.4.3. Supercritical Carbon Dioxide

Supercritical CO_2_ is an important commercial and industrial green solvent due to its low toxicity and does not have a negative environmental impact. Supercritical CO_2_ exists at temperatures above 31 °C and pressures above 73.8 bar; therefore, its handling requires special high-pressure equipment. However, due to the efficiency of the ongoing reactions and the ease of separating the solvent from the products, supercritical CO_2_ is widely used in industry. Below are some recent examples of the use of supercritical CO_2_ as a solvent in enzymatic polymerization reactions.

The preparation of biodegradable polyglycerol succinate, a promising biorenewable surfactant, was carried out by polycondensation in supercritical CO_2_ in the presence of Novozym 435. This method succeeded in obtaining a polymer with a low degree of branching, while carrying out the reaction without a solvent leads to highly branched, crosslinked and insoluble materials [241]. Biocompatible polyester for biomedical purposes, poly-ω-pentadecalactone, was prepared by ring-opening polymerization in supercritical CO_2_ in the presence of Novozym 435 at 200 bar and 70 °C for 2 h. At a ratio of ω-pentadecalactone to CO_2_ of 2:1, the yield in the polymerization reaction was about 60%, and the average molecular weight of the polymer reached 33 kDa [99]. Under similar conditions with similar parameters, poly-ε-caprolactone was obtained, which is also an important biocompatible material [100].

## 6. Integration of Bio- and Chemocatalysis

The analysis of publications related to enzyme-catalyzed polymer synthesis demonstrates that role of biocatalysis in this area is rapidly developing, especially in the production of polymers for biomedical applications. At the same time, the range of polymers obtained by enzymatic polymerization can be significantly expanded due to the integration of biocatalysis and chemical synthesis. A combination of the advantages of chemical and enzymatic methods can be a powerful tool in polymer chemistry for obtaining a wide range of functional polymeric materials, including those for use in medicine [242]. Chemical polymerization of oligomers obtained by enzymatic polymerization can increase the molecular weight and the degree of crystallinity of the final product. So, for example, biodegradable polybutylene succinate was produced in two steps. At the first stage, enzymatic polycondensation of 1,4-butanediol and diethyl succinate using CALB in isooctane at 50 °C was performed, and then the temperature was raised to 90 °C to carry out post-polymerization in bulk (without solvent) [101]. Another example demonstrates the preparation of cytocompatible materials based on polyglycerol fragments, which can potentially be used for controlled release of drugs, in two consecutive steps: enzymatic polymerization using CALB followed by chemical modification to impart desired properties, such as amphiphilicity [243,244,245].

Recent work demonstrated a very interesting scheme for the chemoenzymatic preparation of thermosensitive polymers based on star-like polyesters [73]. The described method included biocatalytic synthesis of monomeric lactone followed by its chemical copolymerization with a previously obtained star-like polycaprolactone, biocatalytic modification of the copolymer and, finally, the chemical formation of crosslinks, which have the property of breaking down on heating and forming on cooling (Figure 17).

At the first stage, norcamphor lactone (NCL) was synthesized with almost 100% conversion from the natural compound norcamphor (NC) using the engineered variant of cyclo-hexanone monooxygenase from *A. calcoaceticus* (CHMO_Acineto_) as a catalyst. Then, the activated monomeric residue was incorporated in the synthesis of block star-like copoly-mers. Star-like polycaprolactone (PCL-4OH) was obtained by the chemical ring-opening polymerization of ε-caprolactone (ε-CL) using di-trimethylolpropane (Di-TMP), which has four primary hydroxyl groups, as an initiator, in the presence of 1.5.7-triazabicyclo[4.4.0]dec-5-ene (TBD). The authors report that the molecular weight of each branch of PCL-4OH was 2650 Da, which corresponds to a degree of polymerization of about 23. The copolymerization of PCL-4OH with NCL was carried out in the presence of methanesulfonic acid (MSA) in toluene, while the initiators of NCL ring-opening polymerization were terminal hydroxyl polymer groups PCL-4OH. A star-like copolymer PCL-B-PNCL-4OH was obtained, with the molecular weight of each branch doubling. Finally, the resulting copolymer was functionalized with furan moieties by biocatalytic transesterification with immobilized lipase B from *Candida antarctica*. Thermoresponsive networks were generated via the reaction with 1,1’-(methylenedi-4,1-phenylene)-bis-maleimide (BMI) by Diels–Alder (DA) chemistry. The Diels–Alder cycloaddition reaction is reversible with temperature changes; thus, the bonding/debonding state of the designed gels can be modulated in response to a specific thermal treatment [73].

## 7. Biocatalytic Production of Monomers

To carry out enzymatic polymerization reactions, the appropriate monomers are required. In the ideal case, meeting the requirements of “green chemistry” when obtaining biodegradable polymers, ‘renewable’ monomers from natural sources should be used. Often, hydroxy acids, amino acids, carbohydrates and other natural compounds cannot be directly introduced into the enzymatic polymerization reaction but must be activated or modified accordingly. Several examples of enzymatic and chemoenzymatic synthesis of monomers suitable for the preparation of biodegradable polymers have recently been published. The ring-opening polymerization is a convenient method for the synthesis of polydepsipeptides and their copolymers using lipases, as mentioned above; however, for its application, available methods for the preparation of monomers, namely 2,5-diketomorpholine derivatives, are needed [246]. A simple two-step method of stereoselective chemoenzymatic synthesis of 2,5-diketomorpholine derivatives was described, which consists in the penicillin acylase-catalyzed formation of N-(hydroxyacyl)-amino acid derivative from the amide of hydroxy acid and amino acid followed by activation of carboxyl group with N,N’-dicyclohexylcarbodiimide and cyclization (Figure 18) [247]. This method makes it possible to obtain a wide range of enantiomerically pure derivatives of 2,5-diketomorpholine with good yield of 80–90% and use them for further enzymatic polymerization to obtain biodegradable and biocompatible polyethers of different structure and variable physicochemical characteristics.

2,5-Furandicarboxylic acid (FDCA, Figure 19) is a promising bio-based building block as a green alternative to petroleum-based terephthalate in polymer production. Polyesters based on FDCA are a new class of biobased polymers of great interest, both from scientific and industrial perspectives [248,249]. Most FDCA is produced by the oxidation of 5-hydroxymethylfurfural (HMF, Figure 19) derived from hexose [250]. Although the chemical conversion is widely applied, the biocatalytic conversion is expected to be competitive due to the relatively mild condition and consumption of fewer toxic chemicals [251]. However, it is difficult to catalyze the conversion of HMF to FDCA using a single enzyme. An enzymatic cascade reaction leading to a product yield of 94% was introduced by the combining 5-hydroxymethylfurfural oxidase (HMFO) of *Methylovorus* sp. MP688 and Novozym 435 [252].

An efficient and highly selective biocatalytic approach for the synthesis of FDCA from 5-hydroxymethylfurfural (HMF) was successfully developed using a 2,2,6,6-tetramethylpiperidine-1-oxyl (TEMPO)/laccase system coupled with *Pseudomonas putida* KT2440. TEMPO/laccase afforded the selective oxidation of the hydroxymethyl group of HMF to form 5-formyl-2-furancarboxylic acid as a major product, which was subsequently oxidized to FDCA by *P. putida* KT2440. Manipulating the reaction conditions resulted in a good conversion of HMF (100%) and an excellent selectivity of FDCA (100%) at substrate concentrations up to 150 mM within 50 h [253]. The cascade catalytic process established in this and previous works offers an ecologically attractive approach to produce FDCA.

A novel whole-cell biocatalyst was constructed by coexpressing vanillin dehydrogenase and HMF/furfural oxidoreductase (HmfH) in *Escherichia coli* for cascade catalytic oxidation of HMF to 2-furancarboxylic acid under sacrificial substrate-free conditions. Under pH-controlled conditions, the biocatalyst enabled the efficient synthesis of FDCA from 150 mM HMF in a 96% yield. Besides, FDCA was prepared on the gram scale with a productivity of around 0.4 g/L·h [254]. It is important to note that oxidoreductases used to catalyze the synthesis of FDCA require cofactors, such as NAD^+^/NADH. It would be too expensive to add an equimolar amount of cofactor to the system; therefore, systems to regenerate cofactors were used, including whole-cell ones [255] or photo-activated ones based on quantum dots [256].

Cadaverine (α,ε-pentamethylenediamine) has been proposed as a promising substitute of 1,6-diaminohexane that can polymerize with dicarboxylic acids to form biobased polyamides. The catalytic form of the lysine decarboxylase was prepared by immobilization and mutation to synthesize cadaverine from lysine [257,258,259].

A chemoenzymatic method was also proposed for the preparation of enantiomerically pure substituted lactones used as monomers for the synthesis of chiral polyesters. The key stage of this methodology was the acylation of racemic substituted homoallylic alcohols followed by stereoselective hydrolysis catalyzed by wild and mutant forms of lipase B from *Candida antarctica*. Different lactone monomers were obtained in six stages, one being biocatalytic and leading to the enantiomeric excess of stereoselective hydrolyses 90–99% with a conversion of 42–99% [260].

Along with the known biocatalytic method for obtaining lactones from the corresponding cyclic ketones using Baeyer–Villiger monooxygenases [261], a method for obtaining ε-caprolactone from cyclohexanone according to Baeyer–Villiger was developed, modified by the addition of a stage of biocatalytic generation of peroxyacetic acid as an oxidizing agent of cyclohexanone. The cross-linked immobilized lipase from *Trichosporon laibacchii* catalyzed the oxidation reaction of ethyl acetate with a complex of urea and hydrogen peroxide to form peroxyacetic acid, ethanol and oxygen with a reported 98.1% conversion of cyclohexanone. In addition, the same enzyme, when toluene was added to the reaction mixture, catalyzed the further polymerization of ε-caprolactone with a yield of 92.4% [262].

An interesting example of the chemoenzymatic synthesis of macromonomers for the production of environmentally friendly epoxy resins was described. The authors suggested using unsaturated fatty acids obtained from vegetable oils as starting substrates. At the first stage, transesterification of triglycerides of fatty acids was carried out to obtain methyl esters of fatty acids followed by epoxidation of double bonds. At the second stage, Novozym 435-catalyzed transesterification of methyl esters of epoxidized fatty acids into glycol ethers was carried out to obtain branched monomers. Alcohols containing two, three or four hydroxyl groups were used as nucleophiles. Finally, at the third stage, crosslinks were formed between the branched monomers by opening epoxy rings upon heating or irradiation or in the presence of polyamines. The resulting material with crosslinked polymer chains had similar properties to hardened epoxy resins [263].

## 8. Conclusions/Perspective

Analysis of the accumulated experience concerning the use of enzymes in polymer chemistry shows that the number of polymers synthesized using enzymes has been significantly expanded in recent years, and new niches have been found. Moreover, it has been shown that biocatalysis provides mild polymerization conditions, which are especially important for the production of labile monomers. The prospects of biocatalysis in obtaining previously unavailable monomers for polymerization were shown. Enzymes can be used to obtain polymers of regular structure when high stereo-, regio- and chemoselectivity of reactions is required. The use of enzymes can help in the production of new complex functional polymers based on natural compounds such as hydroxy and amino acids and renewable monomers from biomass such as 2,5-furandicarboxylic acid, cadaverine, triglycerides of fatty acids and unsaturated fatty acids. The use of enzymes makes it possible to replace chemical catalysts based on organometallic compounds, which is especially important in the production of biomedical materials, where the content of heavy metals is strictly limited. Further research should uncover the industrial potential of biocatalysis for polymer production. Research so far is mainly focused on studying the possibilities of using enzymes in polymer chemistry, while for the technological implementation of scientific developments, it is necessary to solve a whole range of problems: from the choice of the starting material and the optimal form of the biocatalyst, conducting kinetic studies to find optimal conditions for the biocatalytic process, to the reactor design and isolation of the target product. There is interest in the study of various aspects of the use of biocatalysis in this area. However, studies related to substrate, medium and protein engineering; the production of biocatalysts; and reactor technology are still very limited. Not enough attention is paid to the low rate and degree of polymerization in enzyme-catalyzed reactions, as well as to the separation of the biocatalyst from the products, while the mass fraction of the biocatalyst in the reaction mixture can reach 10%. The mechanical incorporation of the enzyme into the product structure can be a problem and also affect the properties of the polymeric material. Adequate mathematical modeling of the entire process using supercomputers can be recommended for finding optimal conditions and integrating various stages of biocatalysis engineering up to the isolation of the product of interest.

Enzymatic polymerization, along with its documented advantages, has numerous limitations. The enzymatic polymerization seems to be less universal compared to chemical methods and needs careful optimization for each specific case (temperature, pH, solvent, composition of the reaction mixture, nature and form of the biocatalyst); however, this should still be studied in detail. With an increase in the degree of polymerization, the viscosity of the reaction mixture increases, which requires novel technological solutions. The advantage of enzymes operating under mild reaction conditions can be considered as a certain limitation, especially when higher temperatures are needed because of the physical characteristics of substrates and products. The products of enzymatic polymerization are most often polymers, the number average molecular weight of which rarely exceeds 30-40 kDa. Such oligomeric blocks can be either modified by chemical methods or used, for example, in the preparation of block copolymers. Therefore, the integration of bio- and chemocatalysis seems to be one of the key directions for the nearest future and can become a powerful tool in polymer chemistry for obtaining more diverse and functional polymeric materials, including those for use in medicine.

## Figures and Tables

**Figure 1 molecules-26-02750-f001:**
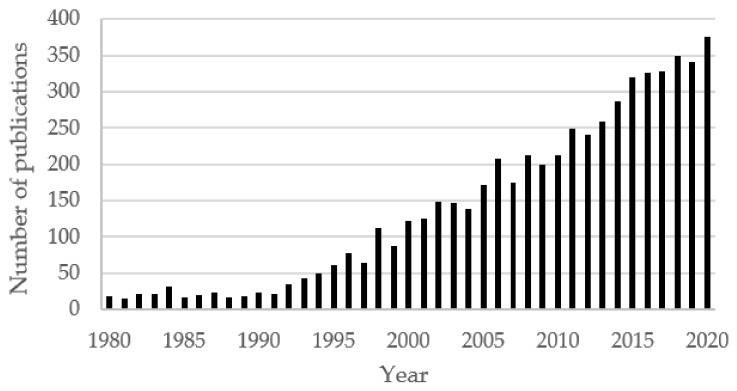
Number of publications versus years by search query “enzymatic polymerization” (by www.scopus.com, accessed on 01 March 2021).

**Figure 2 molecules-26-02750-f002:**
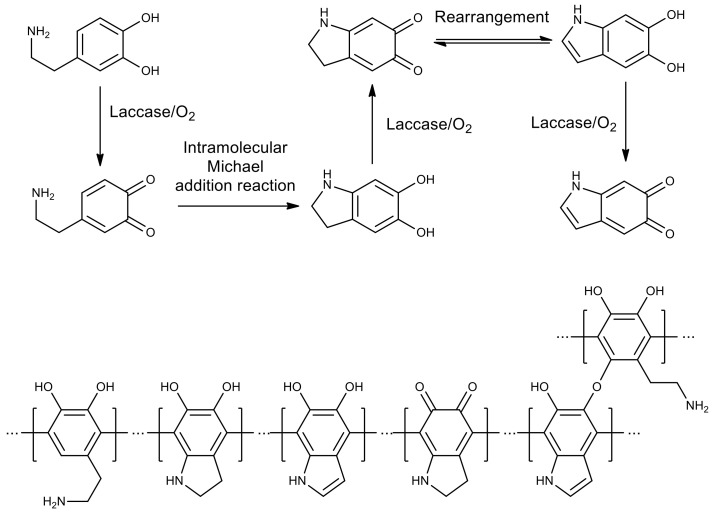
Scheme of the formation of monomers during the oxidation of dopamine with molecular oxygen in the presence of laccase and the structure of the polymer chain of polydopamine, which includes several types of monomer units and additional ether bonds between the chains.

**Figure 3 molecules-26-02750-f003:**
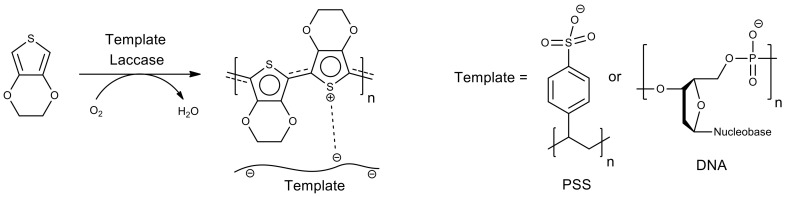
Schematic illustration of the laccase-catalyzed 3,4-ethylenedioxythiophene polymerization in the presence of polystyrene sulfonate (PSS) or DNA as a template.

**Figure 4 molecules-26-02750-f004:**
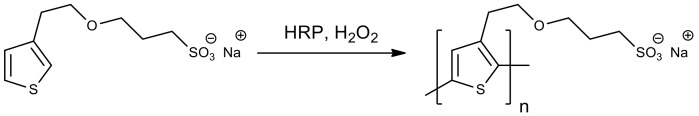
Scheme of enzymatic polymerization of water-soluble (3-thienyl)-ethoxy-4-butyl sulfonate in the presence of horseradish peroxidase (HRP).

**Figure 5 molecules-26-02750-f005:**
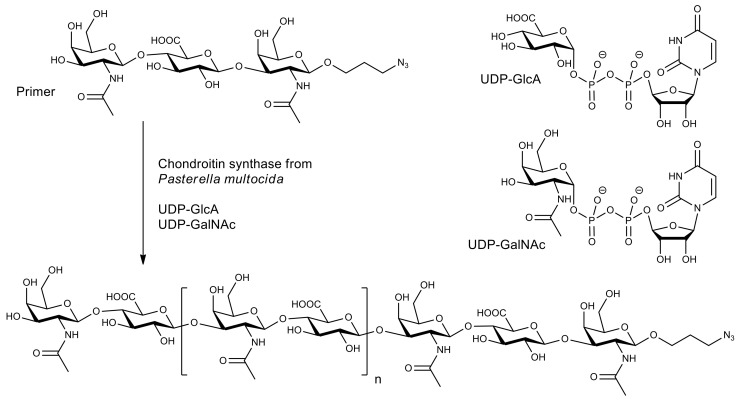
Scheme of biocatalytic preparation of a chondroitin polymer from a trisaccharide primer and monomers UDP-GlcA and UDP-GalNAc.

**Figure 6 molecules-26-02750-f006:**
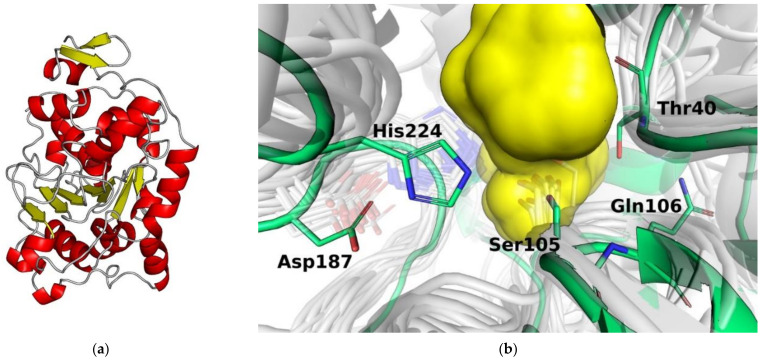
(**a**) Structure of the α/β-hydrolase motif of lipase B from *Candida antarctica*; (**b**) *Candida antarctica* lipase B active site (bold lines) compared to α/β-hydrolase family enzymes, including lipases and proteases (transparent lines). The view is taken from the inner side of catalytic amino acid residues shown by sticks; substrate-binding pocket of lipase B from *Candida antarctica* is shown by yellow surface. The figure was prepared using PyMol based on the crystal structure PDB ID: 1TCA.

**Figure 7 molecules-26-02750-f007:**
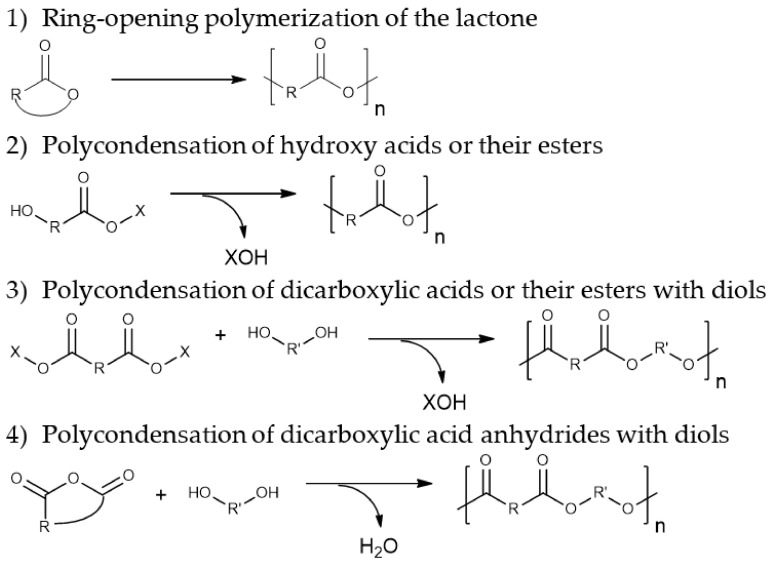
Methods for obtaining polyesters. R, Rʹ—substituents in the molecules of dicarboxylic acid and diol; X—H, Me, Et.

**Figure 8 molecules-26-02750-f008:**
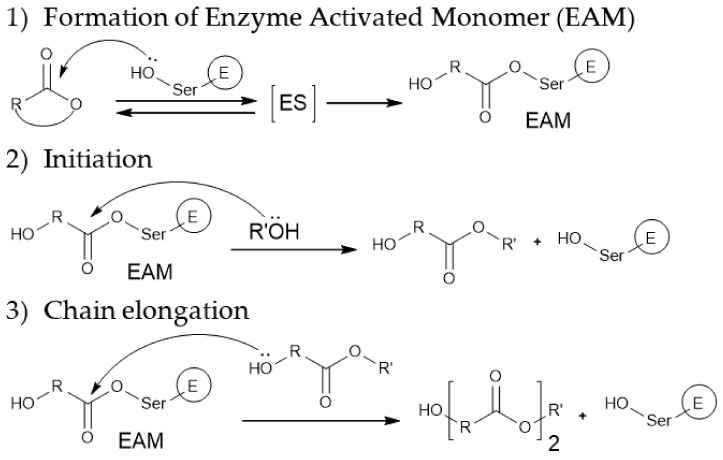
Lipase-catalyzed lactone polymerization mechanism. R represents a group of the lactone molecule; Rʹ—H, alkyl.

**Figure 9 molecules-26-02750-f009:**
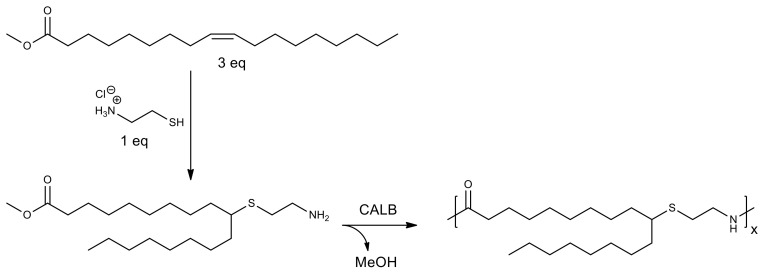
Chemoenzymatic synthesis of polyamide based on a monomer obtained by radical addition of cysteamine to methyl oleate.

**Figure 10 molecules-26-02750-f010:**
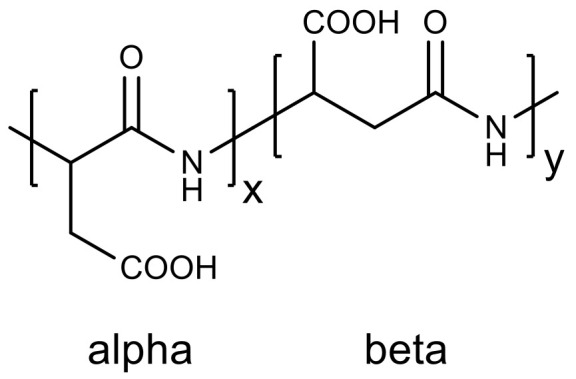
The structure of polyaspartic acid with alpha- and beta-units.

**Figure 11 molecules-26-02750-f011:**
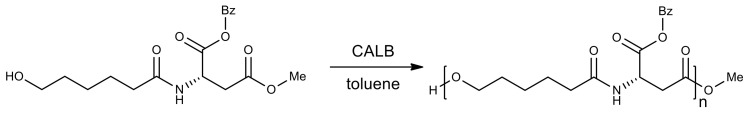
CALB-catalyzed polymerization of N-(6-hydroxyhexanoyl)-L-aspartate.

**Figure 12 molecules-26-02750-f012:**
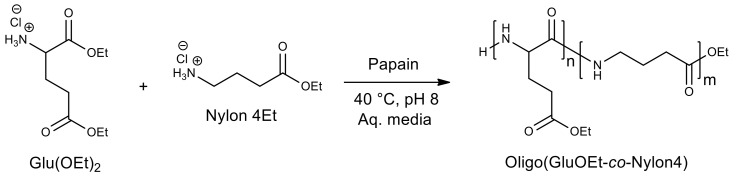
Synthesis of a copolymer of glutamic acid ethyl ester and nylon 4 using papain.

**Figure 13 molecules-26-02750-f013:**
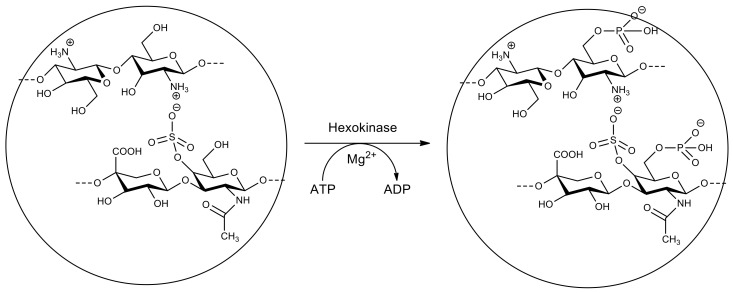
Phosphorylation of nanoparticles consisting of a chitosan–chondroitin sulfate complex using hexokinase.

**Figure 14 molecules-26-02750-f014:**
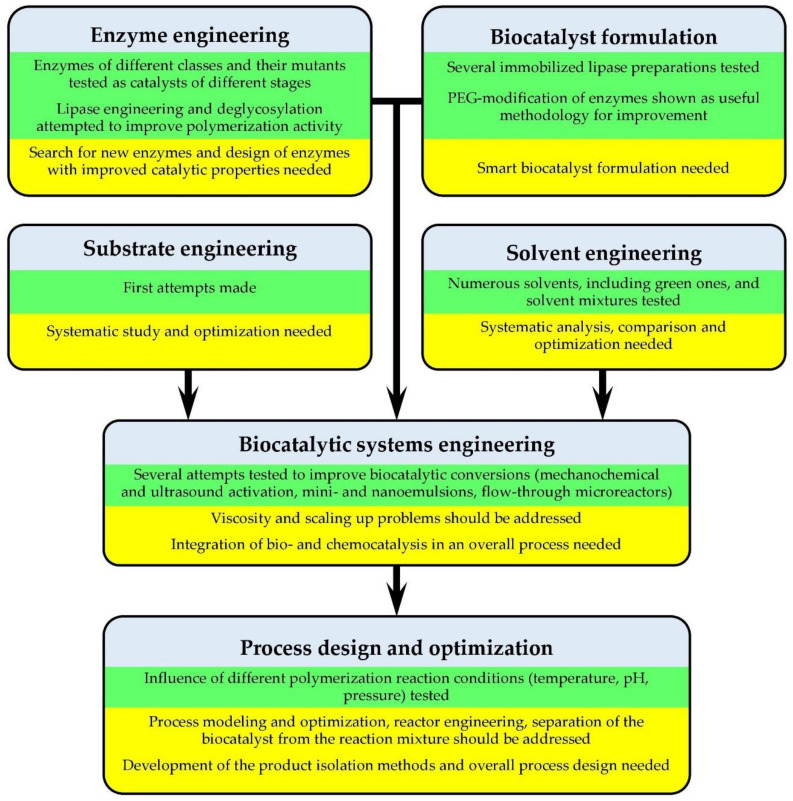
Development of various aspects of the biocatalysis engineering concept for the production of polymers using enzymes: state of the art 2021. Existing developments are highlighted green, problems to be solved are highlighted yellow.

**Figure 15 molecules-26-02750-f015:**
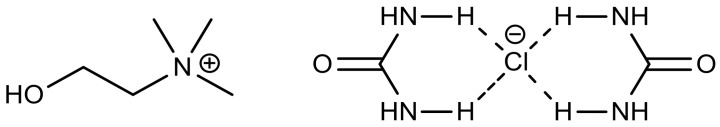
Scheme of the complex formation of a hydrogen bond donor (urea) with a chloride anion in a deep eutectic solvent based on a mixture of urea and choline chloride in a 2:1 molar ratio.

**Figure 16 molecules-26-02750-f016:**
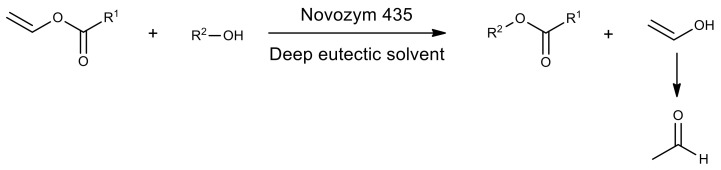
Scheme of the transesterification reaction between vinyl esters and alcohol. R^1^ = n-C_11_H_23_, R^2^ = _H_-C_4_H_9_, _H_-C_8_H_17_, _H_-C_18_H_37_.

**Figure 17 molecules-26-02750-f017:**
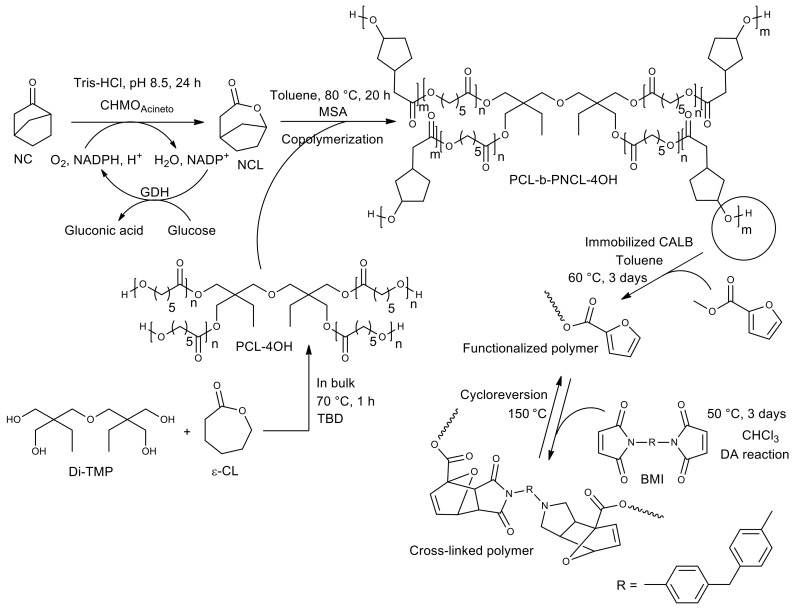
Chemoenzymatic synthesis of thermosensitive polymers based on star-like polyesters. NC—norcamphor, NCL—norcamphor lactone, CHMOAcineto—cyclohexanone monooxygenase from *A. calcoaceticus*, GDH—glucose dehydrogenase, Di-TMP—di-trimethylolpropane, ε-CL—ε-caprolactone, TBD—1,5,7-triazabicyclo[4.4.0]dec-5-ene, MSA—methanesulfonic acid, PCL—polycaprolactone, PNCL—poly(norcamphor lactone), CALB—lipase B from *Candida antarctica*, BMI—1,1′-(methylenedi-4,1-phenylene)-bis-maleimide, DA—Diels–Alder.

**Figure 18 molecules-26-02750-f018:**
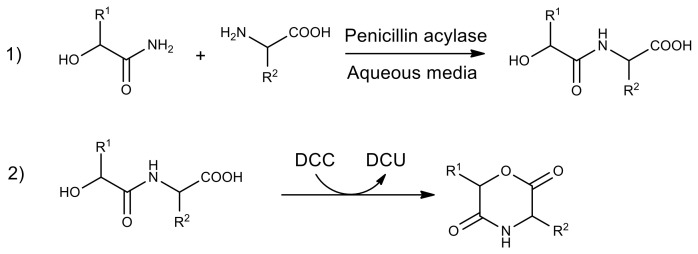
Chemoenzymatic synthesis of 2,5-diketomorpholine derivatives. R^1^—hydroxy acid side chain radical, R^2^—amino acid side chain radical, DCC—N,N’-dicyclohexylcarbodiimide, DCU—N,N’-dicyclohexylurea.

**Figure 19 molecules-26-02750-f019:**
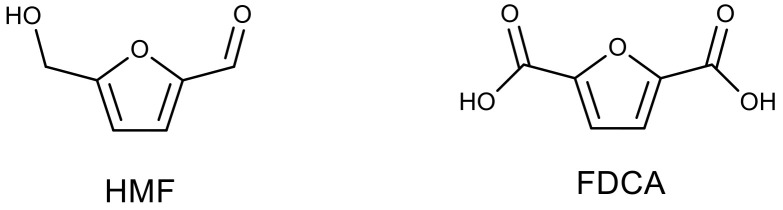
The structures of 2,5-furandicarboxylic acid (FDCA) and 5-hydroxymethylfurfural (HMF).

**Table 1 molecules-26-02750-t001:** Classification of plastics according to [46].

	Bioplastics (from Renewable Resources)	Plastics Made from Petroleum (from Fossil Resources)
Biodegradable plastics	polyoxyacids (poly-lactic acid) polyhydroxyalkanoates polysaccharides (unsubstituted) polyamino acids polyesteramides polydepsipeptides	poly-ε-caprolactone poly(butylene succinate-adipate) poly(butylene adipate-terephthalate)
Non-biodegradable plastics	substituted polysaccharide derivatives polyol-polyurethane bio-polyethylenebio bio-poly(ethylene terephthalate)	polyethylene polypropylene polystyrene poly(ethylene terephthalate)

**Table 2 molecules-26-02750-t002:** Enzymes and polymers produced with them.

Enzyme Class	Enzyme	Polymer Type	Features	References
Oxidoreductases	Laccases	Polyanilines and polyphenols	Aqueous solution, pH 3–6, 30 °C, atmospheric air	[59,60,61,62,63,64]
		Polythiophenes		[65,66,67]
	Peroxidases	Polyphenols	Aqueous micellar system, H_2_O_2_ is added by small portions throughout the entire process	[68]
		Polythiophenes		[69]
	Baeyer–Villiger monooxygenases (BVMO)	Lactones—monomers of polyesters	Aqueous solution, pH 8.5, atmospheric air, NADPH, H^+^ cyclic regeneration system based on alcohol dehydrogenase- or glucose oxidase-catalyzed reactions	[70,71,72,73]
Transferases	Chondroitin synthase (glycosyltransferase)	Polysaccharides (chondroitin)	UDP-GlcA and UDP-GalNAc required as monomers, a regeneration system is needed	[74]
	Polyhydroxybutyrate synthase (acyltransferase)	Poly-3-hydroxybutyrate	Propionyl-CoA transferase for regeneration of coenzyme A is needed	[75,76]
Hydrolases	Lipases	Aromatic and aliphatic polyesters	The most commonly immobilized CALB (Novozym 435) is used, organic solvents (e.g., toluene), ionic liquids, including deep eutectic solvents, supercritical CO_2_ or solvent-free (in bulk), 50–100 °C	[73,77,78,79,80,81,82,83,84,85,86,87,88,89,90,91,92,93,94,95,96,97,98,99,100,101]
		Polyamides and poly(amino acids)		[102,103,104,105,106]
		Polyesteramides		[107,108,109,110]
	Cutinases	Polyesters based on adipic acid and 1,4-butanediol	Immobilized cutinase 1 from *Thermobifida cellulosilytica*, solvent-free, 70 °C	[111,112,113]
	Papain	Oligopeptides and oligopeptidomimetics	Aqueous solution, pH 8-9, 40 °C	[114,115,116,117,118,119]
	Proteinase K	Cysteine oligopeptide		[120]

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
