# Peer review of "Prospects of Using Biocatalysis for the Synthesis and Modification of Polymers"

_molecules, 2021, doi:10.3390/molecules26092750_

Round 1

Reviewer 1 Report

The current work is really a well-written and well-structured review paper and to my opinion it can be published in its present form. It gives the most important current aspects on enzymatic polymerization using also a critical perspective. I specially acknowledge the necessity of the chapter “biocatalysis engineering” which is mandatory for processes scaling up.

I would like to make a suggestion to the authors: to construct a table with the most important polymer repeating units prepared so far via enzymatic polymerization along with some reaction conditions (e.g. temperature, time, solvent) and the relevant reference, so the author can direct himself to the correct reference number to search more for his preferred polymer.

Author Response

We would like to thank the reviewer for kind evaluation and remarks that helped to improve our manuscript. Corrections made and answers to every remark are indicated below.

The current work is really a well-written and well-structured review paper and to my opinion it can be published in its present form. It gives the most important current aspects on enzymatic polymerization using also a critical perspective. I specially acknowledge the necessity of the chapter “biocatalysis engineering” which is mandatory for processes scaling up.

Thanks a lot for supporting this idea.

I would like to make a suggestion to the authors: to construct a table with the most important polymer repeating units prepared so far via enzymatic polymerization along with some reaction conditions (e.g. temperature, time, solvent) and the relevant reference, so the author can direct himself to the correct reference number to search more for his preferred polymer.

Thanks for this suggestion. Made accordingly.

Reviewer 2 Report

The manuscript by Nikulin et al. discuss the use of biocatalyst in modication, depoymerization, degradation, and synthesis of differemt polymers. As the authors mentioned, the reviews on polymers and enzymes have been distinguished; the authors have covered quite wide range of subjects with long breath such as degradation, enzymatic polymerization, biocatalytic modification, biocatalyst design and engineering process, combination of biocatalyst with normal chemical catalyst, and monomer synthesis with more than 250 references. I would like to value their thorough review on very recent reports, about 70% of the references are in 5 years, which is great. It also is good to see that, on cited works, the authors made comments on what remaining problems are and how that would be resolved. Here are some suggestions: in section 3, it would be good to have Table showing the enzymes, its functions, polymerization types, and so on, so that the readers can get a sense in a easy way. In section 5, only texts are found without any figures, thus it is quite tough to followe. I would suggest including at least three Figures that help to follow the contents. Also, a Table would be helpful to sum up the contents for the general readerships. With further improvements, I believe the manuscript will be publishable in Molecules.

Minor comment

Typo: Line 70 Chem-ical; Line 151 poly-mers; Figure 3 chemical structure of PSS is wrong.

Author Response

We would like to thank the reviewer for kind evaluation and remarks that helped to improve our manuscript. Corrections made and answers to every remark are indicated below.

The manuscript by Nikulin et al. discuss the use of biocatalyst in modication, depoymerization, degradation, and synthesis of differemt polymers. As the authors mentioned, the reviews on polymers and enzymes have been distinguished; the authors have covered quite wide range of subjects with long breath such as degradation, enzymatic polymerization, biocatalytic modification, biocatalyst design and engineering process, combination of biocatalyst with normal chemical catalyst, and monomer synthesis with more than 250 references. I would like to value their thorough review on very recent reports, about 70% of the references are in 5 years, which is great. It also is good to see that, on cited works, the authors made comments on what remaining problems are and how that would be resolved.

Thank you very much for these comments.

Here are some suggestions: in section 3, it would be good to have Table showing the enzymes, its functions, polymerization types, and so on, so that the readers can get a sense in a easy way.

Thank you for this suggestion. Made accordingly.

In section 5, only texts are found without any figures, thus it is quite tough to followe. I would suggest including at least three Figures that help to follow the contents. Also, a Table would be helpful to sum up the contents for the general readerships. With further improvements, I believe the manuscript will be publishable in Molecules.

Thank you for this suggestion. Added summarizing illustrations (Figure 14) and comments in the text.

Minor comment

Typo: Line 70 Chem-ical; Line 151 poly-mers; Figure 3 chemical structure of PSS is wrong.

Corrected